Individual and combined ecotoxic effects of water-soluble polymers

Hisar Olcay Hisar@em.uni-frankfurt.de ohisar74@gmail.com 1
Oehlmann Jörg 1 2
1 Department Aquatic Ecotoxicology, Goethe University , Frankfurt am Main , Hessen , Germany
2 Kompetenzzentrum Wasser , Frankfurt am Main , Hessen , Germany
Sotelo-Mundo Rogerio
Electronic publication date: 2023 Nov 23
Publication date: 2023
Volume: 11
Electronic Location ID: e16475
Received 2023 Jun 26; Accepted 2023 Oct 26
Copyright: ©2023 Hisar and Oehlmann
Copyright year: 2023
Copyright holder: Hisar and Oehlmann
License: This is an open access article distributed under the terms of the Creative Commons Attribution License, which permits unrestricted use, distribution, reproduction and adaptation in any medium and for any purpose provided that it is properly attributed. For attribution, the original author(s), title, publication source (PeerJ) and either DOI or URL of the article must be cited.
License URL: https://creativecommons.org/licenses/by/4.0/

Keywords: Mixture toxicology, Concentration addition, Independent action, In vitro assays, Raphidocelis subcapitata

Funding: Johann Wolfgang Goethe-University Frankfurt This work was supported by the Philipp Schwartz Initiative of the Alexander von Humboldt Foundation in the frame of Johann Wolfgang Goethe-University Frankfurt awarded Philipp Schwartz fellowship. The funders had no role in study design, data collection and analysis, decision to publish, or preparation of the manuscript.

==============================
Water-soluble polymers (WSPs) are a class of high-molecular-weight compounds which are widely used in several applications, including water treatment, food processing, and pharmaceuticals. Therefore, they pose a potential threat for water resources and aquatic ecosystems. We assessed the ecotoxicity of four WSPs—non-ionic polyacrylamide (PAM) and polyethylene glycol (PEG-200), anionic homopolymer of acrylic acid (P-AA), and cationic polyquaternium-6 (PQ-6)—as single compounds and in mixture. For this purpose in vitro and in vivo assays were used to record baseline toxicity, mutagenic potential, endocrine effects, and growth inhibition in the freshwater alga Raphidocelis subcapitata. Furthermore, the mixture toxicity of the two polymers P-AA and PQ-6 which showed effects in the algae tests was evaluated with the concentration addition (CA), independent action (IA), and generalized concentration addition (GCA) model and compared with experimental data. No toxic effects were observed among the polymers and their mixtures in the in vitro assays. On the contrary, in the growth inhibition test with R. subcapitata the cationic PQ-6 caused high inhibition while the anionic P-AA and its mixture with the cationic polymer caused low inhibition. The non-ionic polymers PEG-200 and PAM showed no effect in R. subcapitata in the tested concentration range up to 100 mg/L. The IA model represented the mixture effect of the combination experiment better than the CA and GCA models. The results indicate (1) that the toxic effects of anionic and cationic polymers are most likely due to interactions of the polymers with the surfaces of organisms or with nutrients in the water and (2) that the polymers elicit their effects through different mechanisms of action that do not interact with each other.

Introduction

Synthetic polymers are formed from a variety of monomers, which are small molecules that are chemically linked together to form larger polymer chains. The properties of the resulting polymer depend on the specific monomers and the way they are linked together (Nguyen & Kausch, 1999; ECHA, 2023). Water-soluble synthetic polymers (WSPs) have a wide range of properties and applications, including as thickeners, emulsifiers, and dispersants in personal care products and cosmetics (Huppertsberg et al., 2020; Rozman & Kalčíková, 2021), detergents (DeLeo et al., 2020a), as flocculants in wastewater treatment plants (Sgroi et al., 2014) and as soil conditioners to improve water retention and reduce erosion in agricultural fields (Weston et al., 2009).

Solid polymers have received more attention in recent years compared to WSPs (Xiong et al., 2018; Arp & Knutsen, 2020). This is likely due to the visible and widespread presence of the solid polymers in the environment, particularly in marine and freshwater ecosystems (Llorca et al., 2020; Ziajahromi et al., 2020). However, this should not lead to neglecting the potential environmental impact of WSPs because they are discharged into the sewer system or even directly released into the environment (Julinová, Vaňharová & Jurča, 2018; Kawecki & Nowack, 2019; Arp & Knutsen, 2020).

WSPs are considered as being of low environmental concern due to their generally high molecular weight (MW) (Duis, Junker & Coors, 2021), although the environmental persistency of certain WSPs like polyacrylamides and polycarboxylate is well established (Arp & Knutsen, 2020). However, the environmental impact can vary depending on the polymer properties such as MW, charge and the percentage of low MW components and water properties (e.g., water hardness, dissolved organic carbon) (OECD, 2009; Simões et al., 2022). In support of this assumption, ecotoxicological studies have shown that non-ionic polymers are non-toxic, while anionic WSPs show low toxic effects on aquatic organisms. Studies with cationic polymers have reported higher toxicity to aquatic organisms compared to non-ionic and anionic polymers (DeLeo et al., 2020b; Duis, Junker & Coors, 2021).

Nevertheless, aquatic organisms in their natural habitat are typically not exposed to single chemicals but rather to complex mixtures. The mixture toxicity of chemicals can be higher than the effects of its individual components and these mixtures may produce a significant effect. Therefore making such a judgment without considering the mixture toxicity may lead to false assumptions in environmental risk assessments (Backhaus et al., 2000).

Since it is impossible to test every possible combination of individual substances, models have been developed that can be used to predict the combined effects of complex mixtures of substances. Concentration addition (CA) and independent action (IA) are two commonly used models for predicting the combined effects of mixtures of chemicals (Olmstead & LeBlanc, 2005). CA assumes that the individual chemicals in the mixture act together in a concentration-additive manner. In other words, the combined effect of the mixture is equal to the sum of the effects of the individual chemicals, weighted by their respective concentrations. CA is applicable when the chemicals in the mixture share the same mode of action. IA assumes that the individual chemicals in the mixture act independently of each other. The combined effect of the mixture is equal to the product of the effects of the individual chemicals, each raised to the power of its concentration in the mixture. IA is applicable when the chemicals in the mixture have different modes of action (Cedergreen et al., 2008; Howard & Webster, 2009; Altenburger et al., 2013). The generalized concentration addition (GCA) model is an extension and generalization of the concept of CA (Howard & Webster, 2009). The most obvious feature of the GCA approach is that it conceptually divides the cumulative effect of all substances in a mixture into two main parts, and considers together the effect associated with the accumulation of each substance and the effect associated with the mixture of different substances (Tanaka & Tada, 2017).

In our study, four selected WSPs—non-ionic polyethylene glycol (PEG-200) and polyacrylamide (PAM), anionic homopolymer of acrylic acid (P-AA), and cationic polyquaternium-6 (PQ-6)—were first tested individually with a broad range of in vitro assays to account for different modes of action. Furthermore, the single compounds and their binary mixtures were analyzed for in vivo effects in an algae growth inhibition test to provide robust data for an environmental risk assessment. In addition, the CA, IA, and GCA models were applied for mixtures of single polymers which showed an inhibitory effect in the bioassay and the model predictions were then compared with the experimentally determined mixture data.

Materials & Methods

Test chemicals

The four WSPs used as test compounds for the single substance tests and as mixture components in our study were purchased from Sigma-Aldrich (Schnelldorf, Germany): polyacrylamide (PAM; CAS No: 9003-05-8; average MW: 10,000 Da), polyethylene glycol (PEG-20; CAS No: 25322-68-3; average MW: 200 Da); anionic homopolymer of acrylic acid (P-AA; CAS No: 9003-04-7; average MW: 2,100 Da); cationic polyquaternium-6 (PQ-6; CAS No: 26062-79-3, average MW: 100,000 Da). The chemical properties and structures of the WSPs are given in Table 1. Generally, water solubility, charge and MW are the polymer properties considered most relevant with regard to the environmental fate and effects of polymers (Duis, Junker & Coors, 2021). In the selection of the WSPs, attention was paid to obtain polymers with the lowest MW belonging to the non-ionic, anionic and cationic polymer classes. The test compounds were dissolved in water and stored at 4 °C. The same stock solutions were used throughout the study. A spacing factor of 2 was used to delineate the concentration of the WSPs in the media used for the test (e.g., 1.5, 3, 6, 12, 25, 50, 100, 200 mg/L). All tests were performed using at least six different concentrations. The concentration range tested was generally 0.07 to 100 mg/L, but a range of 0.01 to 0.24 was used when preliminary data for the particular WSP indicated a higher toxicity, while a range of 100 to 1,000 mg/L was used when a lower toxicity was observed.

Table 1 Chemical properties and structures of the water-soluble polymers used as test compounds. n.a. = not available.

Polymer class	Technical name	Systematic name	Average molecular weight and density	Chemical structure	
Polyacrylamides	Polyacrylamide (PAM)	poly(2-propenamide)	10,000 Da n.a.		
Polyethylene glycols	Polyethylene glycol(PEG-200)	Poly(oxy-1,2-ethanediyl), α-hydro- ω-hydroxy-	200 Da 1.16 g/mL		
Polycarboxylates: polyacrylates	Anionic homopolymer of acrylic acid(P-AA)	2-Propenoic acid, homopolymer, sodium salt	2,100 Da 0.55 g/mL		
Polyquaterniums	Cationic polyquaternium-6(PQ-6)	2-Propen-1-aminium, N,N-dimethyl-N-2-propen-1-yl-, chloride (1:1), homopolymer	100,000 Da 1.09 g/mL		

Microtox assay

The Microtox assay with the bioluminescent bacterium Aliivibrio fischeri assessed the baseline toxicity (ISO-Guideline 11348-3, 2017). The assay was performed in a miniaturized 96-well plate format according to Völker et al. (2017). Luminescence was measured prior to and 30 min after sample addition using a Spark 10 M microplate reader (Tecan, Crailsheim, Germany). In total, three independent experiments were conducted for every single compound and their binary mixtures in the concentration range between 7.81 and 1,000 mg/L with 8 concentrations and a spacing factor of 2.

Ames fluctuation assay

The Ames fluctuation assay was applied to evaluate the mutagenic potential of the WSPs used in the study. The assay was performed according to Magdeburg et al. (2014) in accordance with ISO guideline 11350 (ISO-Guideline 11350, 2012) with Salmonella typhimurium strains YG 1041 and YG 1042 (Hagiwara et al., 1993) with and without metabolic activation by rodent liver enzymes (S9-mix; Envigo CRS, Roßdorf, Germany). Mutagenicity was photometrically determined by a colour change of the indicator bromocresol purple (CAS: 115-40-2; Merck, Darmstadt, Germany) at 420 nm (Spark M; Tecan, Crailsheim, Germany). Wells with an optical density of >0.45 (and a clear colour change) were considered mutagenic. For every single compound and their binary mixtures, triplicate experiments were conducted as a limit test at a concentration of 1,000 mg/L.

Yeast-based reporter gene assays for endocrine and dioxin-like activities

Yeast-based reporter gene (lacZ encoding for β-galactosidase) assays were conducted to assess the endocrine and dioxin-like activity. We examined the agonistic activities at the human estrogen receptor α (hERα) (Yeast Estrogen Screen = YES; Routledge & Sumpter, 1996) and human androgen receptor (hAR) (Yeast Androgen Screen = YES; Sohoni & Sumpter, 1998) as well as the activation of the human aryl hydrocarbon receptor (Yeast Dioxin Screen = YDS; Miller, 1997) according to the protocol of Abbas et al. (2019). Activities were photometrically determined at 540 nm (Multiskan Ascent, Thermo Labsystems) through cleavage of chlorophenol red- β-D-galactopyranoside (CAS: 99792-79-7; Sigma-Aldrich, Steinheim, Germany) by β-galactosidase over a period up to 60 min at 5-10 min intervals. In total, three independent experiments were conducted for every single compound and their binary mixtures as a limit test at a concentration of 1,000 mg/L.

72-h growth inhibition assay with Raphidocelis subcapitata

The 72 h growth inhibition experiment was performed using R. subcapitata algae in the exponential growth phase. This experiment was carried out according to OECD guideline 201 adapted to 24-well microplates (Moreira-Santos, Soares & Ribeiro, 2004). The algae were exposed to six concentrations of each polymer (for PAM, PEG-200, and P-AA ranging from 3.2 to 100 mg/L; for PQ-6 ranging from 0.015 to 0.48 mg/L), a positive control (potassium chromate at a concentration of 8.262 ×10−9 mol/microplate-well) and a negative control for 96 h. Three replicates were performed for each concentration and control with two mL of test solution and a cell density of 20,000 cells/mL. The experiments were carried out under continuous illumination (4 klx/m2/s), at 26 ± 1 °C, 60% humidity. To prevent cell clumping, an automatic stirrer at 220 rpm was used. Mixture tests were carried out by mixing polymers which showed inhibitory effects on growth in R. subcapitata. Nine concentrations were prepared for each polymer mixture with 5 replicates by following the procedure described above. Absorbance (ABS) values were measured at 595 nm after 24, 48, 72 and 96 h of exposure using a Tecan microplate reader (Tecan, Crailsheim, Germany). After each measurement, plates were sealed with Breathe-Easy-membrane.

Statistical analysis and mathematical modelling

GraphPad Prism (versions 5 and 8; GraphPad Software, San Diego, CA, USA) were used for statistical and nonlinear regression analyses of the data obtained from the experiments. One-way analysis of variance (ANOVA) followed by Dunnett’s test was applied to analyse significant differences among treatments at the p < 0.05 significance level.

Concentration-response curve fits and EC50 values with 95% confidence intervals were calculated only for chemicals that showed significant effects in the R. subcapitata growth inhibition test. The top value was set to the data point with the greatest inhibitory effect in the test. Top values and EC50 values were used to predict mixture toxicity with the models in an Excel spreadsheet.

Equation (1) was used for CA modelling. (1) ECxmix=∑i=1npiECxi−1

where ECxmix is the total concentration of the mixture at which x% effect occurs and pi is the fraction of component i in the mixture. ECxi is the concentration of i mixture component that provokes x% effect when applied singly.

Equation (2) was used to calculate the mixture effects according to IA. The concentration–response relationships Fi of the individual components were used to calculate their effects E(ci) as (2) Ecmix=1−∏i=1n1−Fci

If the concentrations of the individual components are expressed as fractions of the total concentration, the overall effect of any given total mixture concentration can be calculated as according to Eq. (3) (3) Ecmix=1−∏i=1n1−Fpi∗cmix

where pi is the fraction of component i in the total mixture concentration and ∏ stands for the product (multiplication).

Equation (4) was used to calculate the mixture effects according to GCA. (4) E=max effect levelAAEC50A+max effect levelBBEC50B+…1+AEC50A+BEC50B+…

In Eq. (4), ‘E’ represents the effect of the mixture at the given concentration of the chemicals A ([A]) and B ([B]). The term ‘max effect level’ stands for the maximum effect levels of the chemicals A and B as individual compounds, respectively, while ‘EC50A’ and ‘EC50B’ represent their median effect concentrations as individual compounds. Using Eq. (4), concentration-response curves were calculated for various concentrations of the two compounds in binary mixtures.

Results

In vitro assays

None of the four individual substances and none of their binary mixtures showed statistically significant effects in the investigated concentration range in the in vitro assays. In the Microtox assay to assess baseline toxicity, the maximum inhibition of luminescence was only 1.4% and 5.5% for PAM and PEG-200, respectively, at the highest test concentration of 1,000 mg/L. These inhibitions were not statistically significantly different from the control (one-way ANOVA, p > 0.05). Likewise, none of the other WSPs or their mixture showed an inhibition of luminescence in the Microtox assay.

The results of the Ames assay with strains YG 1041 and YG1042, both with and without S9 mix, also provided no evidence of mutagenic activity of the tested individual WSPs or their binary mixtures at a concentration of 1,000 mg/L, since the revertant proportions remained well below the required threshold of 20.8% for a positive test outcome according to ISO-Guideline 11350 (2012). Revertant proportions above 10% were only observed in two tests: 14.0% for PEG-200 with strain YG 1041 without S9 and 15.8% for the PEG-200/P-AA mixture with strain YG 1042 without S9, while the revertant proportions for all other individual WSPs and their binary mixtures were below 10%. In assays with S9 mixture, no revertants appeared for any of the substances or mixtures.

The agonistic activities determined at the estrogen receptor α, the testosterone and aryl hydrocarbon receptors were well below the limit of detection (LOD) of the respective yeast-based reporter gene assays (LOD: YES = 0.002 µg 17 β-estradiol equivalents/L; YAS = 0.176 µg testosterone equivalents/L; YDS = 11.2 µg β-naphthoflavone equivalents/L). Accordingly, even at the tested concentration of 1,000 mg/L, the results do not indicate any receptor mediated estrogenic, androgenic or dioxin-like activity for the individual WSPs and their binary mixtures.

72-h growth inhibition assay with Raphidocelis subcapitata

In contrast, P-AA at concentrations of 25, 50 and 100 mg/L and PQ-6 at concentrations of 0.05 and 0.1 mg/L as single substances caused a statistically significant (p < 0.05 significance level) reduction of specific growth rate in the microalga R. subcapitata, but no toxicity effects were detected for the non-ionic WSPs PEG-200 and PAM (Fig. 1 and Fig. 2A). The toxic effect of P-AA in binary mixtures with PAM and PEG-200 on algal growth was the same as that of P-AA alone, but the effect of PQ-6 in binary mixtures with the non-ionic WSPs on algal growth was only observed when PQ-6 was at the highest concentration (0.1 mg/L). Among the binary mixtures, the mixture of P-AA and PQ-6 with a concentration ratio of 1/1,000 had the highest toxic effect (Fig. 2B).

Figure 1 Growth curves of Raphidocelis subcapitata cultures under control conditions and under exposure to the four tested water-soluble polymers polyacrylamide (PAM), polyethylene glycol (PEG-200), anionic homopolymer of acrylic acid (P-AA) and cationic polyquaternium-6 (PQ-6) over a period of 96 h.

Figure 2 Growth rates of Raphidocelis subcapitata under exposure to single compounds (A) and binary mixtures (B) of the four tested water-soluble polymers.

Mixture ratio is provided in parentheses in the figure legend of (B) for the four tested water-soluble polymers polyacrylamide (PAM), polyethylene glycol (PEG-200), anionic homopolymer of acrylic acid (P-AA) and cationic polyquaternium-6 (PQ-6) in the growth inhibition test after 72 h. * Significantly different from the control, p < 0.05.

P-AA, PQ-6 and their mixture exerted different toxicities to R. subcapitata with EC50 values after 72 h ranging from 0.04 to 35.5 mg/L (95% confidence limits). PQ-6 showed to be more toxic with a 72 h EC50 (95% CL) of 0.06 (0.04–0.08) mg/L than P-AA with 24.2 (16.5–35.5) mg/L. The 72 h EC50 value determined for their mixture (9.74 mg/L; 95% CL 8.06–11.8 mg/L) is lower than for P-AA and higher than for PQ-6 (Table 2 and Fig. 3).

Table 2 Key parameters of calculated concentration response curves for water-soluble polymers in the growth inhibition test with Raphidocelis subcapitata after 72 h.

Parameters are provided for PAM, PEG-200, P-AA and PQ-6 as single compounds and as binary mixture of P-AA and PQ-6. Furthermore, the key parameters predicted by the concentration addition (CA), generalized concentration addition (GCA), and independent action (IA) models are provided.

Components	EC50	95% confidence intervals for EC50 (mg/L)	Top value (Max. effect level in %)	Hillslope	
PAM	–	–	–	–	
PEG-200	–	–	–	–	
P-AA	24.2 mg/L = 11.5 µM/L	16.5 to 35.5	94	1.41	
PQ-6	0.06 mg/L = 6 × 10−4 µM/L	0.04 to 0.08	100	1.16	
Binary mixture(P-AA+PQ-6)	9.74 mg/L = 4.64 µM/L	8.06 to 11.8	95	1.57	
CA prediction	4.68 mg/L = 2.23 µM/L	4.60 to 4.77	100	1.15	
GCA prediction	7.10 mg/L = 3.38 µM/L	6.49 to 7.76	100	0.99	
IA prediction	9.28 mg/L = 4.42 µM/L	8.65 to 9.96	100	1.10	

Figure 3 Concentration-response curves for growth inhibition in Raphidocelis subcapitata under exposure to the cationic polyquaternium-6 (PQ-6) and the anionic homopolymer of acrylic acid (P-AA) as single compounds and their binary mixture at a PQ-6 to P-AA ratio of 1 to 1,000 after 72 h.

The broken line represents the 50% inhibition level. EC50 values are provided in Table 2.

The results of the experimental determination of the mixture toxicity of P-AA and PQ-6 as well as the predictions made by CA, IA, and GCA concepts are shown in Fig. 4. The CA model predicts a higher mixture toxicity than the IA and GCA models. The observed EC50 value of the mixture was 9.74 mg/L, while the values estimated by CA, GCA and IA were 4.68 (95% CL 4.60–4.77), 7.10 (6.49–7.76) and 9.28 (8.65–9.96) mg/L, respectively. Accordingly, only the IA model exhibited overlapping 95% confidence limits of the calculated EC50 value with the experimentally determined EC50.

Figure 4 Comparison of concentration–response curves for growth inhibition in Raphidocelis subcapitata for a binary mixture of PQ-6 and P-AA after 72 h as determined experimentally and predicted by the concentration addition (CA), generalized concentration addition (GCA), and independent action (IA) models.

Discussion

In our study, reported effect concentrations are based on nominal concentrations without an analytical verification of the actual concentration in the test vessels because specific methods for the chemical analysis of WSPs are mostly lacking (ECETOC, 2020; Duis, Junker & Coors, 2021; Pauelsen et al., 2023).

The results of the in vitro assays carried out did not provide any indication of a measurable baseline toxic, endocrine, dioxin-like or mutagenic effect of the four test substances or their binary mixtures in the considered concentration range of up to 1,000 mg/L. This is in line with the findings of other studies, which determined little (acute LC50 or EC50 10–100 mg/L; chronic NOEC or EC10 1–10 mg/L) or no ecotoxicity (acute LC50 or EC50 >100 mg/L; chronic NOEC or EC10 >10 mg/L) according to United Nations (2015) for the majority of the WSPs investigated to date (Wagner, Nabholz & Kent, 1995; Duis, Junker & Coors, 2021). The low toxicity of the substances is mostly attributed to their molecular size, which limits the uptake via membranes into cells and biota (Hamilton, Reinert & Freeman, 1994; ECETOC, 2020). The passage of polymers with a MW exceeding 1,000 Da through membranes has been considered unlikely (Boethling & Nabholz, 1997). In yeast and bacteria, the cell wall is a further factor limiting the uptake of WSPs into the cell (Aouida et al., 2003). In addition, the bacteria used in the Microtox and Ames fluctuation assay are Gram-negative bacteria, whose surface can bind cationic WSPs much better than Gram-positive bacteria due to the seven times higher negative charge density (Wilhelm et al., 2021). This hinders also the passage of anionic WSPs.

Most non-ionic polymers are generally considered to be biocompatible and non-toxic (Nabholz & Zeeman, 1991; Harford et al., 2011). In our algal studies, similarly, PEG-200 and PAM with MWs of 200 and 10,000 Da, respectively had EC50 values > 100 mg/L, indicating a lack of toxicity according to United Nations (2015) (Table 2). However, it should be noted that in many cases, the sum of the concentrations of WSP-related by-products and transformation products in the effluent of wastewater treatment plants may exceed the concentrations of the precursors (Freeling et al., 2019). Therefore, the environmental maximum concentrations of WSPs should be taken into account for environmental risk assessments. On the other hand, algae can detect extracellular signals that allow them to survive and thrive in certain environments. These signals can come from abiotic environmental factors, neighboring cells, or other organisms. When algae sense these signals, they activate intracellular pathways to adapt to the new conditions. These pathways involve a series of molecular events that allow algae to adapt their physiology and behavior, accordingly, including gene expression changes and modulations of cell size. This also allows the algae to adapt to unfavorable environmental conditions within certain limits (Nayaka, Toppo & Verma, 2017). The tested P-AA as a representative of anionic WSPs in our study has a mean MW of 2,100 Da, which indicates a low uptake across membranes. The EC50 in the growth inhibition test with R. subcapitata was 24.2 mg/L, so this substance has a low ecotoxicity according to United Nations (2015). This is in line with the conclusion of DeLeo et al. (2020a) that P-AAs have low ecotoxicity and do not pose significant risks to aquatic organisms. However, the ecotoxicity of P-AAs is influenced by factors such as MW, structure, concentration, charge, expose duration and water hardness (Hamilton, Reinert & Freeman, 1994). Boethling & Nabholz (1997) reported EC50 values for P-AAs with MW between 1.4 and 78 kDa ranging from 3.13 to > 100 mg/L and chronic effect concentrations from 0.5 to > 100 mg/L, demonstrating the dominating influence of the MW. They also reported that P-AAs with very high oligomer content (40% and 49%) had the lowest EC50 values. In particular, the highest toxicity was observed in standard algal growing media with low water hardness (10–24 mg/L as CaCO3). This indicates that P-AA may have an indirect toxic effect on algae growth due to the chelation of cationic trace elements (Ca2+, Mg2+ and Fe3+), similar to the previous assessment of Boethling & Nabholz (1997).

In ecotoxicological studies, cationic polymers have a higher toxic effect on aquatic organisms than anionic and non-ionic polymers (Bolto & Gregory, 2007; Duis, Junker & Coors, 2021). Generally, algae are expected to be very sensitive to polyquaterniums and other cationic polymers (NICNAS, 2009). They affect algal growth, photosynthesis, and membrane integrity by their electrostatic interactions with the negatively charged membranes of the algae (Nolte et al., 2017). For PQ-6, an EC50 of 0.06 mg/L was determined in our tests with R. subcapitata. Salinas et al. (2020) and Duis, Junker & Coors (2021) reported EC50 values of 0.03 and 0.16 mg/L for PQ-6, 0.04–0.05 mg/L for PQ-10 and between 0.1 and 1.10 mg/L for PQ-16. The effects might be attributed to negatively charged pectin, a polyanionic polysaccharide, in the outer cell wall (Haas et al., 2021), since pectins appear to play a critical role in stress relaxation during growth, cell-wall integrity, detection of plant pathogens, and defence response (Wang, Kanyuka & Papp-Rupar, 2022).

The component-based approaches such as IA, CA, and GCA allow to estimate the impact of mixtures of chemicals without the need for additional data on mixture toxicities unless there are strong interactions (synergistic or compensatory) between component chemicals (Cedergreen et al., 2008; Backhaus & Faust, 2012). Therefore, in this study, we also investigated how accurately the models CA, IA and GCA predicted the combined effect of P-AA and PQ-6 observed in the growth inhibition test with R. subcapitata. In Fig. 4, the IA model yielded a curve that met a larger range of the concentration–response curve as compared to the CA and GCA models. Also, IA was the only model that resulted in overlapping 95% confidence limits of the calculated EC50 value compared to the experimentally determined. This may be due to a different mode of action of P-AA and PQ-6 in the test with the microalgae (Table 2 and Fig. 3), while substances with similar modes of action tend to follow the CA model (Altenburger et al., 2000; Backhaus et al., 2000). However, there were some differences between the observed and predicted values in the concentrations above and below the EC50 of the mixture. These differences may be explained by the fact that the response shape for a mixture depends on the transformed concentrations rather than simply on the fractions of the components (Tanaka & Tada, 2017). We suppose that P-AA with its steeper response curve (Hillslope 1.41) has a more dominating influence on the response curve of the mixture than PQ-6 (Hillslope 1.16). This is supported by the Hillslope of 1.57 in the mixture which is closer to that of P-AA than PQ-6 (Table 2 and Fig. 3).

Conclusions

The non-ionic WSPs PAM and PEG-200 did not cause any acute toxicity in the eco-toxicity assays, but long-term studies at lower concentrations, also considering multiple stressors, are needed as non-ionic WSPs are more easily biodegraded than P-AAs and PQs, which are slowly biodegradable (Duis, Junker & Coors, 2021). Since the toxic effect of WSPs to aquatic organisms is related to the polymer structure, it was hypothesized that oligomer and monomer structures of non-ionic WSPs may have different toxic behaviour as a result of biodegradation (Arp & Knutsen, 2020). In contrast to the non-ionic polymers, the cationic PQ-6 exhibited a high toxicity, while the anionic P-AA had a low toxicity in the test with R. subcapitata. Considering that any potential toxic effects on the algae may cause damage to organisms at higher trophic levels, the impact of WSPs on algae requires further attention. In our study we also found that the IA model provided the best prediction compared to the CA and GCA models. The CA concept overestimated the observed mixture toxicity by a factor of 2.1, the GCA model by a factor of 1.4. However, the inaccuracy of the CA and especially of the GCA model is relatively small.

Supplemental Information

Supplemental Information 1 Polyacrylamide (PA), polyethylene glycol (PEG-200), anionic homopolymer of acrylic acid (P-AA) and cationic polyquaterniums (PQ-6) tested in the growth rate and inhibition tests with R. subcapitata after 72 h at different concentrations

Click here for additional data file.

Additional Information and Declarations

Competing Interests

Author Contributions

Data Availability

Jörg Oehlmann is an Academic Editor for PeerJ.

Olcay Hisar conceived and designed the experiments, performed the experiments, analyzed the data, prepared figures and/or tables, authored or reviewed drafts of the article, and approved the final draft.

Jörg Oehlmann conceived and designed the experiments, performed the experiments, analyzed the data, prepared figures and/or tables, authored or reviewed drafts of the article, and approved the final draft.

The following information was supplied regarding data availability:

The raw measurements are available in the Supplementary File.

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
