# Peer review of "Individual and combined ecotoxic effects of water-soluble polymers"

_PeerJ, doi:10.7717/peerj.16475_

## Round 0.1 · original submission · Major Revisions

Could you kindly provide a thoroughly revised version that takes into account the editorial feedback and also include a detailed rebuttal letter? Your efforts are much appreciated.

Reviewer 1 ·

Basic reporting

As per the Title which explains the research intent - The article is fully aligned with it.
The article focuses on Water-soluble polymers (WSPs) a class of high-molecular-weight compounds which are highly mobile and potentially threaten water resources and aquatic ecosystems. As these are widely used in various applications, including water treatment, food processing, and pharmaceuticals, the ecotoxicity of four WSPs - non-ionic polyethylene glycol (PEG-200) and polyacrylamide (PA), anionic homo-polymer of acrylic acid (P-AA), and cationic polyquaterniums (PQ-6) - as single compounds and in the mixture were examined for in vitro and in vivo assays for its toxicity.

English used in the manuscript was very professional.
Literature references were of the required field.

Experimental design

The experimental design and intentions were very well explained and all the activities were given in detail.
The effects on the algae tests were evaluated for these water-soluble polymers with the concentration addition (CA), independent action (IA), and generalized concentration addition (GCA) model and compared with experimental data.

They reported that P-AAs with very high oligomer content (40% and 49%) had the lowest EC50 values. The highest toxicity was observed in standard algal growing media with low water hardness (10-24 mg/L as CaCO3). This indicates that P-AA may have an indirect toxic effect on algae growth due to the chelation of cationic trace elements (Ca2+, Mg2+, and Fe3+).

Methods were well defined and sufficient details were provided.

Validity of the findings

Though the novelty is missing the manuscript describes the toxic ecological effect due to water soluble polymers with different combinations. The finding reflects that the non-ionic WSPs PA and PEG-200 did not cause any acute toxicity in the eco-toxicity assays.WSP are easily biodegradable. The findings reflect that cationic PQ-6 exhibited high toxicity, while the anionic P-AA had low toxicity in the test with R. subcapitata.

The experimental findings were very well stated and all the data (assay) were provided.
Conlusion is well satted.

Additional comments

The research article is very well-defined and justifies the title of the research.

Reviewer 2 ·

Basic reporting

Figures must be improved; they need more resolution and are not appropriately described and labeled.
The chemical structure of the polymers used must be included in the manuscript.

Experimental design

Additional information on polymers stock solutions (solvent used, specific polymer concentration, density or viscosity, zeta potential) is needed.
It is necessary to include the growth curve of the microalgae Raphidocelis subcapitata to understand the changes in microalga culture during the experiments.
Why was the monitoring of pigment and fatty acids composition and photosynthetic performance throughout the experiment not performed? It would reveal a possible dose-dependent response to polymer exposure. In addition, cell morphology observation would reveal possible changes in cell size.

Validity of the findings

R. subcapitata capacity to acclimate under adverse environmental conditions might be related to gene expression changes and cell size reduction. This possibility must be discussed.
Further studies are needed to investigate R. subcapitata changes under multiple stressors, such as stressing light and temperature, during polymer exposure. This perspective might be considered.

Additional comments

Some representative pictures of R. subcapitata cells and culture generated during the experiments would be attractive.

Reviewer 3 ·

Basic reporting

The results of the study are well reported and discussed in a good language with very few minor mistakes. There are two main concerns when reporting and discussing the results:

Molecular weight: The authors correctly identified that the MW may play a major role in the effects for WSPs. In parts of the introduction (e.g. line 51/52) and discussion (for example relating the MW and concentrations used to environmental findings, at least for PEG) it is worth considering that environmental molecular weights of WPS can be different from the ones produced. While studies that address this are still near non-existent a study by Freeling et al (https://www.sciencedirect.com/science/article/abs/pii/S0048969719320030) analysed a selection of individual PEG monomers and Paulsen et al (https://www.sciencedirect.com/science/article/abs/pii/S0048969723021824?via%3Dihub) analysed environmental occurrence and release of PEG with size exclusion chromatography coupled to mass spectrometry.

Molar result reporting: since MW varied drastically between polymers (200-100000 Da) it can be informative to report results not only in mg/L or similar but also in µmol/L or similar.

Experimental design

no comment

Validity of the findings

no comment

Additional comments

Specific comments:
Line 15-16: I would advice against generalizing WSPs as highly mobile since it is largely dependent on the chain lengths. While shorter chain homologues are typically mobile high molecular weight WSPs can readily absorb to soil and sludge.
Line 57 WSPs instead of WCPs.
Line 92-95: WSPs used differ hugely in their molecular weight (200-100000 Da). Please give a brief explanation for the MW selection.

---

## Round 0.2 · accepted · Accept

All reviewer comments have been addressed, and I am pleased to accept the manuscript in PeerJ. Congratulations!

Reviewer 3 ·

Basic reporting

My comments have been addressed adequatly.

Experimental design

My comments have been addressed adequatly.

Validity of the findings

No comments

Additional comments

No comments